# Health Benefits of Dietary Fiber for the Management of Inflammatory Bowel Disease

**DOI:** 10.3390/biomedicines10061242

**Published:** 2022-05-26

**Authors:** Kafayat Yusuf, Subhrajit Saha, Shahid Umar

**Affiliations:** 1Department of Surgery, University of Kansas Medical Center, 3901 Rainbow Blvd, 4028 Wahl Hall East, Kansas City, KS 66160, USA; kyusuf@kumc.edu; 2Department of Radiation Oncology, University of Kansas Medical Center, 3901 Rainbow Blvd, Kansas City, KS 66160, USA; ssaha@kumc.edu

**Keywords:** inflammatory bowel disease, diet, gut microbiome, colon cancer

## Abstract

Crohn’s disease (CD) and ulcerative colitis (UC), two components of inflammatory bowel disease (IBD), are painful conditions that affect children and adults. Despite substantial research, there is no permanent cure for IBD, and patients face an increased risk of colon cancer. Dietary fiber’s health advantages have been thoroughly investigated, and it is recommended for its enormous health benefits. This review article discusses the importance of appropriate fiber intake in managing IBD, emphasizing how optimal fiber consumption can significantly help IBD patients.

## 1. Introduction

Inflammatory bowel disease (IBD) is a severe and painful gastrointestinal tract disease that affects children and adults [1]. IBD is a complex disease that has a detrimental effect on people’s quality of life. Due to the severe inflammation associated with IBD, the symptoms and associated morbidity are always prevalent [2]. Patients with IBD frequently face challenges because the disease is often accompanied by extraintestinal manifestations, is incurable, causes significant morbidity, and increases the risk of colorectal (CRC) cancer [3,4]. The incidence of IBD has also been on the rise globally, particularly in more developed countries and newly industrialized countries [1,2,3,5,6]. Recently published data on the global incidence of IBD revealed that there are currently about 6.8 million patients with IBD globally, with the USA having the highest age-standardized prevalence rate, followed by the UK [2].

There has also been an increase in IBD incidence among the pediatric population [7]. Recent statistical data demonstrated that the pediatric prevalence of IBD in the United States has significantly increased from 33.0 per 100,000 in 2007 to 77.0 per 100,000 in 2016; the increase was particularly prevalent in the 10–17 age subgroups [8].

### 1.1. Classification of IBD

Crohn’s disease (CD) and ulcerative colitis (UC) are the most common subtypes of IBD; however, there is also a minor subgroup known as indeterminate IBD [3,5,6,7].

According to reports, the global prevalence of UC is substantially more significant than the incidence of CD, and it is the most frequent form of IBD [9,10]. Ulcerative colitis seems to have a bimodal distribution, with the age of onset peaking between 15 and 30. The senior population between 50 and 70 also experiences a second incidence wave. Males and females appear to be equally represented in UC, indicating that it is not a gender-biased disease [9,10]. The inflammatory process in ulcerative colitis is often limited to the colon. Diffuse and continuous inflammation extending proximally from the rectum and active neutrophilic inflammation are distinguishing features [1,3,11]. UC appears to damage the superficial mucosa, beginning with the rectum repeatedly, and is limited to the colon, making the symptoms less dynamic [5,12]. Goblet cell loss, lesions, and changes in the mucosal crypts are also signs of UC [10]. Colonic mucosa inflammation in UC patients produces pain, ulcers, hemorrhaging, diarrhea, and electrolyte loss [4,5]. About a third of UC patients experience extraintestinal symptoms, and some individuals had these symptoms before they were diagnosed with IBD [10].

Crohn’s disease is a systemic inflammatory bowel illness that can affect any part of the digestive tract, from the mouth to the anus [13]. CD can present as subversive, lacerating, or stricturing, with some individuals displaying all three phenotypes and the creation of fistulas [1,4,5]. In contrast to UC, the disease severity and location often dictate the associated signs and symptoms, resulting in various clinical presentations [5]. Additionally, the inflammatory pattern associated with CD may facilitate the creation of strictures throughout the gastrointestinal tract, resulting in gastric blockage, colonic obstruction, stomach pain, nausea, severe diarrhea, vomiting, and weight loss [4,5]. Physicians can also use discontinuous inflammation and distinct mouth sores or linear ulcers to separate CD from UC after endoscopy [1]. In contrast to UC, CD has the potential to extend beyond the mucosa, whereas UC is restricted to the mucosal area [1]. Histologic characteristics shared between CD and UC patients include the signs of active inflammation evidenced by an abundance of neutrophils and chronic inflammation delineated by crypt loss or damage, mucin deficiency, and lamina propria lymphocytosis [1]. While cellular and molecular mechanisms of disease have been studied in detail, many independent loci, including NOD2, ATG16L1, STAT3, and IL23 family members associated with innate immune pathways, increase CD risk [14]. Loss-of-function mutations in NOD2 [14,15] stimulate fibrostenotic complications, and are associated with the highest risk of CD [16].

Indeterminate colitis is a poorly characterized condition affecting five to ten percent of people with inflammatory bowel disease [1,3]. Because many CD and UC symptoms are similar, it is challenging for pathologists to discriminate between the two conditions in some patients. Both CD and UC can be present in people with indeterminate IBD, but neither can be reliably diagnosed [1,3,6].

### 1.2. Disease Pathogenesis

Although there is no single cause of IBD, several studies have found a crosstalk between genetic susceptibility, environmental factors, the host-microbiome, and the patient’s immune system that plays a role in the chronicity of the disease [3,12,17]. IBD is caused by various environmental factors, including smoking, diet, medications, location, stressors, and psychological factors [17]. Biologics such as anti-TNF therapy administered to CD patients provide only ~30% mucosal healing [18], while ~40% of patients show no clinical response. More recently, inhibition of JAK-STAT pathways is considered for CD [19] but remain problematic due to side effects [20]. Targeting of IL6 and IL11 for CD treatment is not yet approved while recent efforts have focused on pharmacological therapies targeting integrins in the treatment of IBD and cancer [21].

IBD development and severity have been linked to diet, a potentially modifiable environmental risk factor. A pro-inflammatory diet can cause intestinal inflammation by driving immune system dysregulation, disrupting intestinal permeability and mucus layer, alongside contributing to microbial dysbiosis [22]. As a result, many findings have focused on dietary adjustments as a therapeutic measure for IBD management.

### 1.3. What Is a High-Fiber Diet?

The word “Fiber” does not have a single broadly accepted definition. Fiber is difficult to define because new substances are discovered with chemical properties similar to fiber but not physiological [23]. People eat fiber for its health advantages, and thus a definition based on physiological qualities is often desirable. In all known definitions, fiber is described as “carbohydrate or lignin which bypasses the small intestine and is fermented partially or entirely in the colon” [23,24].

Dietary fiber is composed of indigestible carbohydrates found inherent to and intact in plants. These carbohydrates include the remains of edible plant cells, polysaccharides, lignin, and other compounds that are resistant to digestion by human digestive tract enzymes [24,25,26]. Dietary fiber can consist of pectin, mucilage, resistant starch, lignin, cellulose, hemicellulose, gums, and other related plant substances [25,27]. Diets high in fiber, such as fruits and vegetables, are very beneficial to health because their consumption has been linked to a decrease in the prevalence of numerous diseases [23,27,28].

A high-fiber diet implements a daily dietary fiber intake that equals or exceeds the United States Institute of Medicine’s (IOM) Dietary Reference Intake for dietary fiber [23]. According to a list published by Akbar and Shreenath [23], the current daily recommended dietary fiber consumption ranges between 14 and 20 g for children, 22–30 g for adolescents, and 25–38 g for the elderly.

#### Classification of Dietary Fibers

One standard method for categorizing dietary fiber (Figure 1) is whether it is water-soluble/well-fermented or insoluble/less-fermented [27].

**Soluble fibers**, such as β-Glucans, mucilage, pectin, and gum, are water-soluble fibers generated from the inner flesh of plants. They produce a sticky gel in the colon, where bacteria digest them into gases and by-products such as short-chain fatty acids (SCFAs). Oats, barley, fruits, peas, beans, other legumes, and most root vegetables are high in soluble fibers [23].

Soluble fibers are easily digested and may improve stool volume by supporting the growth of the gut microbiome and its by-products, (SCFAs), which maintain colonic integrity and initiate a cascade of other beneficial effects. These qualities may aid in stool normalization by softening hard stool in constipation and firming loose or watery stool in diarrhea [25].

Pectin: pectin compounds are a diverse collection of polysaccharides primarily composed of d-Galacturonic acid. They are structural components of plant cell walls and serve as intercellular cement. Pectin is highly soluble in water and is digested almost completely by colonic bacteria. These soluble polysaccharides may slow gastric emptying and influence small intestine transit time due to their gelling properties. Due to its particular capacity to form spreadable gels, pectins are used in various jellies and jams in the food business [26,27,29]. Citrus peels and apple pomace are commercial sources of pectin [30].

β-Glucans: polysaccharides called β-Glucans are found in the sub-aleurone and endosperm of cereal seeds such as wheat, oats, and barley. The benefits of consuming β-Glucans include decreasing blood serum cholesterol and regulating blood glucose levels, which may benefit people with diabetes. These health advantages are often linked to the number and molecular weight of solubilized β-Glucans in the gastrointestinal system [30].

Hydrocolloids: soluble dietary fiber includes polysaccharide-based hydrocolloids, which are poorly digestible by human enzymes. Gum arabic, seed gums (galactomannans: locust bean gum, guar gum), fermented gums (e.g., xanthan gum and gellan gum), and seaweed gums are all included in this category [30]. Rather than being part of a plant’s cell wall, gums are a soluble fiber produced in specific secretory plant cells. They are extensively branched water-binding and gel-forming polysaccharides [27].

Mucilages: mucilages are water-soluble fibers found in the outer layer of plantain seeds; they are made up of distinctive mucilage cells. The most popular mucilages are psyllium husk, yellow mustard mucilage, and flaxseed gum. Psyllium husk contains around 70% soluble fibers, also known as psyllium gum. Psyllium gum is a natural laxative that can help to improve colon health and function. Mucilage from yellow mustard is an effective emulsion stabilizer. Flaxseed gum has a low viscosity and is suitable for dairy products [30].

Oligosaccharides: numerous oligosaccharides derived from plants cannot be digested by humans and are grouped as low-molecular-weight soluble dietary fiber. Inulin is a common oligosaccharide that appears as a reserve carbohydrate in various plant groups, including rye, onion, and chicory. Inulin is soluble in warm water, indigestible in the small intestine, but quickly degraded to SCFAs by bacteria in the large intestine. Food industries can use it to substitute for sugar and fat in various food products, including desserts and yogurt [30].

**Insoluble fibers**: insoluble fibers are derived from the outer skin of plants, are less prone to fermentation, are not soluble in water, and, in most cases, cannot be fermented by bacteria in the colon. As an effect, they make up most of the stool and aid in laxation [23,26]. Insoluble fiber sources include cellulose found in potatoes, corn bran, the skin of most fruits, many green vegetables, and some fruit plants.. Whole grains contain hemicellulose, while nuts and seeds contain lignin. All of these are insoluble fibers [23].

Insoluble fibers have a bulking effect of increasing stool mass, relieving constipation, and improving digestion regularity. The increased stool weight is caused by the water held within the fiber matrix. Insoluble fibers are also linked to a reduction in intestinal transit time, which aids in avoiding and relieving constipation [25].

Cellulose: in nature, cellulose is the most dominant carbohydrate; it occurs in conjunction with lignin, pectin, and hemicelluloses [30]. Because it is a crucial component of the cell wall of the majority of plants, it is found in fruits, vegetables, and cereals. It is a polysaccharide composed of up to 10,000 glucose monomer units per molecule; the linear molecules are tightly packed together and are highly insoluble and resistant to digestion by human enzymes. Cellulose accounts for approximately one-fourth of the dietary fiber found in grains and fruit and one-third of the fiber found in vegetables and nuts [25].

As the most well-known type of dietary fiber, cellulose is well-researched and has several health benefits, including reducing constipation and managing weight loss. Due to its availability in plant sources, cellulose is the most prevalent dietary fiber in the mammalian diet and thus the most-investigated type of dietary fiber [29].

Hemicellulose: hemicelluloses are a diverse collection of polysaccharides found in plant cell walls. They contain xylose, pentose sugars, and arabinose, with different sugar units present in varying amounts and substituents. Hemicellulose often occupies the spaces between cellulose fibrils and, in some instances, combines with pectin to form the plant cell wall matrix. Cereals are a good source of hemicelluloses [29,30,31].

Lignin: lignin is found in almost all raw materials used in the food industry. It is made up of polysaccharides that are covalently bonded to phenolic chemicals. Lignin binds with cellulose and hemicelluloses to create lignin–carbohydrate complexes (LCC) in plant tissues, where it forms the plant cell wall. When lignin is attached to polysaccharides, glycosidic bonds, ether bonds, and ester bonds establish a covalent binding between the monolignols of lignin and the monosaccharide residues of the polysaccharide [32]. Lignin is an insoluble dietary fiber that aids in the prevention of bile stone development and cholesterol reduction [29]. Coconut shells, oat husks, rice husks, almond shells, and hazelnut shells are all lignin-containing foods [32].

## 2. High-Fiber Diet for IBD: What We Know

### 2.1. A High-Fiber Diet Can Help Minimize Inflammation

One hallmark of IBD is the persistence of chronic inflammation in the gut of affected patients. Moreover, resolution of gut inflammation in IBD and a complete mucosal healing despite being the ultimate goal remains a therapeutic challenge. Swann and colleagues recently reported that a high-fiber diet might help reduce inflammation by altering the pH and permeability of the gut [33]. Clinical trials on the anti-inflammatory effect of a high fiber diet have also shown promising results in patients with IBD.

In 2011, Benjamin and associates found that patients with active Crohn’s disease who received 15 g of fructo-oligosaccharides (FOS) daily had lower proportions of interleukin (IL-6)-positive lamina propria dendritic cells (DC), higher levels of IL-10 staining in the DC, and no change in IL-12p40 production [34]. Despite affecting DC function, the group concluded that no therapeutic benefit was achieved in patients with active Crohn’s disease; one reason for this could be the amount of dietary fiber, the short trial duration, which lasted only four weeks, and the use of FOS as the only treatment.

Another pilot study evaluated the effect of germinated barley foodstuff (GBF) on serum TNF-α, IL-6, and IL-8 levels in UC patients who were in remission. During the two-month study, patients in the treatment group received 30 g of GBF daily by oral administration and standard drug therapy. TNF-α, IL-6, and IL-8 levels all decreased in the GBF group compared to controls throughout the study [35]. In 2014, the same researchers conducted a follow-up trial to see how (GBF) supplementation affected serum c-reactive protein (CRP) levels and clinical symptoms in patients with UC. According to the findings, serum CRP in the GBF group reduced considerably, and combining GBF with existing medications may effectively reduce inflammation and clinical symptoms in UC patients [36].

A group of researchers also explored a semi-vegetarian diet (SVD), specifically a lacto-ovo-vegetarian diet, for patients with inflammatory bowel disease. The SVD contained 32.4 g of dietary fiber. The study discovered that for newly diagnosed CD patients, the remission rate with combined medication (infliximab) and SVD was 100 percent. At two years, 92% of patients maintained remission on SVD without scheduled maintenance therapy with biological drugs. These excellent outcomes were attributed in part to SVD, which prompted the researchers to recommend a high fiber intake for the treatment of CD [37].

A more recent crossover study of 17 individuals with UC who were either in remission or had minimal disease revealed similar findings. Patients were assigned to a low-fat, high-fiber diet or an improved standard American diet. The results indicated that the low-fat, high-fiber diet decreased indicators of inflammation and dysbiosis [38].

The microbial metabolism of dietary fibers can also lead to the release of ferulic acid (FA) by commensal bacteria [39]. FA exhibits antioxidant and anti-inflammatory characteristics in the digestive tract and could be evaluated as a potential therapeutic treatment for a variety of chronic diseases [39]. FA was found to exert protective anti-inflammatory effects in a trinitrobenzene sulfonic acid (TNBS) induced colitis model. The researchers suggested that FA’s mechanism of action is by regulating oxido-nitrosative stress, apoptosis, pro-inflammatory cytokine release, and COX-2 generation [40].

Additionally, anti-inflammatory properties of high fiber supplementation have been demonstrated extensively using preclinical animal models mimicking human inflammatory bowel disease. These studies discovered that fiber supplements reduced pro-inflammatory markers and repaired the epithelial barrier alongside preventing mucosal damage [41,42,43], (Figure 2).

### 2.2. A High-Fiber Diet Can Help Mitigate Dysbiosis by Restoring the Gut Microbiome

The relationship between the gut microbiome and gut health has been a topic of extensive study in recent times [44,45,46,47]. The human gut microbiota is a distinct habitat that differs significantly from person to person. It is made up of trillions of non-pathogenic microorganisms, most of which are bacterial in origin [44]. The microbiota collaborates with the host’s immune system to prevent disease colonization, gut infiltration from pathogens, and intestinal epithelial damage [48]. Digestion, metabolite synthesis, and immune system conditioning are all aspects of the microbiome that help modify inflammatory processes [49]. The variety and quantity of bacterial species can vary dramatically depending on various circumstances, including health state, with a microbiome with a higher diversity of microorganisms associated with better health [49].

Gut dysbiosis occurs when the composition and function of the gut microbiota are disturbed, resulting in the loss of intestinal homeostasis and inappropriate immune activation [44,50]. Gut dysbiosis has been recognized as a hallmark of IBD, characterized by a loss of microbial diversity, a decline in helpful anaerobic bacteria populations, and an increase in unfavorable adherent and invasive pathogenic bacteria [50]. Patients with inflammatory bowel disease have a decreased diversity of anaerobic bacteria such as Firmicutes but an increased population of Proteobacteria [44,45]. Gut dysbiosis may also damage the intestinal epithelial barrier, promoting heightened immunological responses and persistent inflammation [44]. The ability of commensal gut microbes to degrade fiber into short-chain fatty acids makes a high-fiber diet beneficial for gut microbiome regulation [23,25,30]. SCFAs can include acetate, propionate, and butyrate. They serve as energy sources for colonic epithelial cells and help maintain intestinal homeostasis [44].

Reports have illustrated that a high-fiber diet can transform the microbiome composition of patients with IBD (Figure 2). According to data from a crossover research of UC remission patients who followed a low-fat, high-fiber diet, the relative abundance of Actinobacteria in feces declined following the diet, but the relative abundance of Bacteroidetes increased. After four weeks on the diet, the relative abundance of *Faecalibacterium prausnitzii* was similarly increased. According to the researchers, patients on the diet also had higher fecal levels of SCFA acetate [38]. Another study on CD patients indicated that the fecal bacterial count of *F. prausnitzii* in patients with CD was lower than those of healthy controls [37]. Similarly, Bolte and colleagues discovered that human food patterns could influence the general ability to develop protective gut bacteria or foster an abundance of pathogenic and harmful bacteria [51].

Similar findings have been described in experimental mouse models of colitis. Tian and colleagues demonstrated that insoluble dietary fiber from barley leaves alleviated **Dextran sulfate sodium** (DSS) induced colitis in mice by modulating the gut microbiota. The study found that supplementing with dietary fiber significantly decreased the abundance of Akkermansia and increased the number of Parasutterella, Erysipelatoclostridium, and Alistipes. These effects were eliminated when antibiotics were used to reduce the gut flora [52]. Another DSS-induced colitis model used fermented barley and soybean as a dietary fiber substitute. The study demonstrated that the barley soybean combination reduced epithelial barrier failure, increased tight junction protein levels in colon tissues, and inhibited FITC–dextran permeability. Additionally, the diet boosted Lactobacilli and Bacteroides levels [53]. Furthermore, by rebuilding the microbiome, pectin and tributyrin diets were able to alleviate colitis in mice [54]. Mice fed with pectin showed a decrease in the abundance of Proteobacteria and an increase in the population of Bacteroidetes and Firmicutes [54]. Another study revealed that a lack of dietary fiber alters the gut microbiota composition, promotes neutrophil recruitment, and aggravates colitis in mice [55].

Extensive research into the benefits of healthy gut microbiota has led to the utilization of fecal microbiota transplantations from healthy donors as a therapeutic avenue for patients with IBD [56,57].

### 2.3. A High-Fiber Diet Can Help Modulate Tolerable Immune Response

IBD retains an immune signature that is characterized by a rise in the infiltration of innate immune cells (such as neutrophils, macrophages, and dendritic cells), excessive stimulation of effector T cells, and modified tolerance mechanisms that are mediated by regulatory T cells (Tregs) [58]. The relationship between the gut microbiome, immune system regulation, and its anti-inflammatory capabilities inform us about the ability of dietary fiber intake to facilitate immune system reprogramming in IBD (Figure 2). Several studies have demonstrated that dietary fiber can alter immune responses and have a direct anti-inflammatory effect by interacting directly with pattern recognition receptors, particularly Toll-like receptors in intestinal immune cells [59].

A study examined the effect of apple pectin oligogalactan on (DSS)-induced colitis in mice and discovered that the diet was protective. It acted by lowering LPS-induced TNF-α, which is most likely due to a mechanism involving TLR4 re-localization from the cell membrane to the cytoplasm. The study concluded that apple pectin oligogalactan exerts anti-inflammatory and anti-carcinogenic activity via inhibiting the LPS/TLR4/NF-κB pathway [60]. Also relevant in the context of inflammation is the contribution of TLR activation coupled with pro inflammatory cytokine production and ROS generation that facilitates myeloid-derived cell mobilization from the bone marrow [61]. Indeed, in mouse model of DSS-induced colitis, significant increase in CD14+ monocyte infiltration and its association with fibrotic processes illustrates a therapeutic dilemma for targeting this complex multi-step disease process.

In an elegant study, using interleukin-10 knockout mice with spontaneous colitis, researchers discovered that dietary pectin promoted an anti-inflammatory response. The study revealed that levels of TNF-α and GATA-3 were reduced in the pectin-fed group. Dietary pectin activity was linked to its ability to balance the production of pro-inflammatory cytokines and immunoglobulins, possibly by suppressing Th1 or Th2 immune responses [62].

Sabater and colleagues recently demonstrated that artichoke pectin has anti-inflammatory effects in mice with DSS-induced colitis. The study discovered that mice fed with artichoke pectin had lower pro-inflammatory markers such as TNF-α, ICAM-1, iNOS, and TLR4, implying that artichoke pectin could help alleviate inflammatory bowel disease in a mouse model of colitis [63].

Several other studies have confirmed the involvement of dietary fibers in modifying the innate immune response via interactions with Toll-like receptors and generating an anti-inflammatory response to IBD-related intestinal injury [64,65,66].

High-fiber diets can also modulate adaptive immune response. A recent study on mice demonstrated that high-fiber maternal nutrition during pregnancy and breastfeeding modulates the thymic milieu and induces autoimmune regulator (Aire) expression, a factor in the thymus required for T cell maturation. The maternal fiber intake elevated butyrate levels in the offspring’s blood and contributed to an increase in peripheral and thymic Treg numbers in a GPR41-dependent manner [67].

Fusarawa and colleagues discovered that SCFA butyrate stimulates the differentiation of colonic Treg cells in mice. According to the findings, butyrate reduced the development of colitis in Rag1-/- mice after adoptive transfer of CD4^+^CD45RB^hi^ T cells. Furthermore, butyrate treatment of naive T cells under Treg-cell polarizing circumstances increased histone H3 acetylation in the promoter and conserved non-coding sequence areas of the Foxp3 locus, implying a putative mechanism by which microbial-derived butyrate influences the population of Tregs [68]. In a similar study, researchers discovered that SCFAs could modulate the composition and activity of the colonic Treg pool and protect mice from colitis in a Ffar2-dependent mechanism. According to the findings, microbial metabolites regulate adaptive immune response and microbiome coadaptation, resulting in enhanced colonic homeostasis [69].

Additionally, Yasuma and colleagues recently discovered that ferulic acid or ferulic acid-rich supernatant of a colonic bacterium cultured with insoluble arabinoxylans has anti-inflammatory activity on dendritic cells under inflammatory conditions and enhances the Th1-type immune response in mice under physiological conditions. The researchers concluded that FA produced in the gut by *Bacteroides intestinalis* can safeguard the host’s health by boosting the immune system [70].

### 2.4. A High-Fiber Diet Exerts Its Effects on Cancer Prevention and Therapy

Studies have reported that individuals with IBD have a significantly increased risk of colorectal cancer (CRC), primarily due to the pro-neoplastic effects of chronic intestinal inflammation [71,72,73]. IBD-related colorectal cancer (CRC) accounts for approximately 2 percent of the annual mortality from CRC overall and about 10 to 15 percent of the yearly deaths in IBD patients [72]. Multiple studies have demonstrated the preventive effects of a high-fiber diet on colon cancer [74,75].

In a very comprehensive investigation, Hullings and colleagues discovered the beneficial effects of whole grains as a source of fiber in preventing colorectal cancer. The study employed Cox proportional hazard model to estimate hazard ratios and 95 percent Confidence Intervals for the intake of whole grains and other sources of dietary fiber and the risk of colorectal cancer in over 400,000 US adults aged 50 to 71 over a period of 16 years. The study concluded that whole grains as a source of fiber were inversely correlated to subsites of colorectal cancer, especially rectal cancer [76].

A similar prospective cohort study of 25,000 adults from Alberta, Canada, was established to investigate the link between food habits and cancer incidence. Principle component analysis (PCA) and reduced rank regression (RRR) was employed to determine dietary patterns, while data linkage with the Alberta Cancer Registry was used to assess incident cancer cases. The researchers discovered that a “dietary fiber” pattern reduced the incidence of combination malignancies as well as lung, colon, and prostate cancers. In the same study, a “fructose” pattern was linked to an increased risk of combined cancers and lung cancer, whereas a “discretionary fats” pattern was linked to a lower risk of colon and combined cancers [77].

Another study examined the relationship between dietary fiber consumption and the incidence or recurrence of colorectal cancer. The study discovered that participants with the highest dietary fiber intake, especially from cereals and fruits, had a lower risk of incident colorectal adenoma and distal colon cancer [78]. The study advised that this protective impact of fiber may begin in the early stages of colorectal carcinogenesis [78].

Dietary fiber and fat consumption have also been reported to influence the composition and metabolic function of the gut microbiota. Fiber, in particular, protects against colorectal cancer risk, whereas fat intake increases it [77,78].

These reports suggest that fiber supplementation and fat restriction are promising strategies for lowering CRC risk in healthy individuals (Figure 2). Furthermore, fiber supplementation could serve as an excellent conceptual approach to manipulating microbial metabolism and reducing colon cancer threat in high-risk populations. Fiber supplementation is recommended because high-fat intake influences the bile acid pool and metabolism and must be limited to reduce levels of tumor-promoting deoxycholic acid in the colon [79].

Using the DSS model of colitis-induced cancer, Nishiguchi and colleagues found that a high-fructose diet exacerbated chronic colitis and promoted colitis-associated tumorigenesis. They also discovered that switching the mice to a fiber-rich diet (containing psyllium, pectin, inulin, or cellulose) protected them from chronic colitis and tumorigenesis. At the same time, fiber enrichment restored the animals’ microbiome. Their study discovered that psyllium had the highest efficacy [80].

Investigations with a gnotobiotic mouse model also revealed that dietary fiber could protect against colorectal carcinogenesis. The study demonstrated that fiber has a potent tumor-suppressive effect by utilizing microbiota and butyrate-dependent mechanisms [81].

Interestingly, studies have shown that dietary fiber can help patients respond better to cancer treatment. Song and colleagues conducted a thorough investigation on the link between fiber intake and survival following CRC diagnosis. A total of 1575 individuals with stage I to III colorectal cancer were enrolled in the prospective cohort research. The study discovered that a higher fiber intake, particularly from cereals and whole grains, correlated to a lower risk of CRC progression and overall mortality. The study concluded that patients who increased their fiber consumption after diagnosis compared to pre-diagnostic levels had a higher chance of survival [82].

It is also noteworthy that the benefits of fiber intake during cancer therapy extend beyond CRC. A recent study evaluated the fecal microbiota profiles, dietary patterns, and probiotic supplement use in melanoma patients, alongside conducting similar preclinical studies. The study discovered that in 128 patients on immune checkpoint blockade (ICB), higher dietary fiber consumption was related to a significant increase in progression-free survival. The most significant effect was reported in those with adequate dietary fiber intake and no probiotic usage. In follow-up animal studies, mice fed a low-fiber diet or given probiotics had a decreased response to anti-programmed cell death 1 (anti–PD-1)-based therapy, as indicated by a lower frequency of interferon gamma positive cytotoxic T cells in the tumor microenvironment [83].

These studies highlight the protective benefit of fiber consumption in cancer prevention and its potential to continue to exert its effects amid cancer treatment. These findings offer an excellent foundation for investigating fiber supplementation in combination with existing cancer therapies.

### 2.5. The High-Fiber Diet and Overall Quality of Life

Increased fiber intake from fruits, vegetables, and whole grains has been studied to improve the overall quality of life by promoting a healthy gut (Figure 2). Dietary fiber has been established to influence gastrointestinal tract physiology through several mechanisms [26].

A high-fiber diet can help maintain bowel movements by absorbing water and softening stool, thus preventing the onset or aggravation of hemorrhoids and diverticulitis. Soft feces are easier to pass in this context, preventing constipation. Furthermore, fiber consumption can also add bulk to the stools, preventing the formation of loose stools [23].

Regular fiber consumption can help restructure the digestive system, which is essential for feelings of satiety and food intake control [84]. Fiber can also help modulate digestive processes such as food transit time control, contributing to the regulation of circulating glucose and lipid [26]. In addition, soluble fibers make stomach contents viscous and prolong gastric emptying time, which can help reduce weight, improve BMI, lower body fat, and decrease the waist-to-hip ratio [23]. It should be noted that highly viscous soluble fibers (such as psyllium, β-Glucans, and raw guar gum) have provided this viscosity-dependent health effect [85].

Lastly, dietary fiber consumption has been reported to have the same health benefits for children and adults [86]. Wegh and colleagues conducted a detailed assessment of the effect of fiber and prebiotics on the severity of children’s gastrointestinal diseases and their microbiome composition. According to the review, the microbiome of children with colic and inflammatory bowel syndrome differs significantly from those of the control group [87]. In a similar finding, Gehrig and colleagues described a causal relationship between poor gut microbial composition and childhood malnutrition in a detailed investigation. The researchers discovered a microbiota-directed supplemental food prototype that raised levels of biomarkers and mediators of growth, bone formation, neurodevelopment, and immunological function to levels comparable to those seen in healthy children. The findings imply that a healthy microbiota composition is causally linked to healthy growth and development, suggesting a potential treatment method for childhood undernutrition [88]. 

Health complications such heart disease, stroke, hypertension, diabetes, and other gastrointestinal illnesses such as duodenal ulcer and gastroesophageal reflux disease can also be managed and incidence reduced by increasing fiber consumption [86].

## 3. Mechanism of Dietary Fiber Action: How Does Dietary Fiber Intake Exert Its Effects?

Extensive studies have been conducted on dietary fiber interventions’ impact on modifying the gastrointestinal tract [89,90]. To fully comprehend the overall health advantages of dietary fiber, it is necessary to understand the mechanism by which different forms of dietary fiber work on different individuals and how the makeup of individuals influences the effect they receive from fiber consumption.

### 3.1. Physicochemical Properties of Dietary Fiber (Not All Fiber Are the Same)

Diverse types of fiber have different impacts on the body, and from a functional standpoint, not all fibers have the same benefits; as this relies on the quantity and fiber types [59].

Soluble fibers are highly fermentable, allowing beneficial bacteria to flourish in the colon. Fiber fermentation enables the formation of SCFAs, amines, ammonium, gases, and phenols, which affects the gut microbiota diversity and composition [25,59]. The immune barrier of the GI tract is also preserved by the bloom of beneficial microbiota during fiber degradation in the colon. Fermentation products, primarily SCFAs, interact with the immune cells of the small intestine before being degraded by microbial enzymes, resulting in these positive effects [91].

Due to their inability to retain water, insoluble dietary fibers, like lignin and cellulose, are less fermentable by the gut microbiome [59]. Insoluble dietary fibers have a fiber matrix that holds water, increases stool mass, alleviates constipation, and improves bowel regularity by providing a bulking effect. Additionally, insoluble fibers have been linked to a reduction in intestinal transit time, which can help avoid and alleviate constipation [59]

Psyllium and other gel-forming fibers thicken the contents of the intestinal lumen and impede the migration of nutrients to the intestinal walls. As a result, they can lower cholesterol, sugar, and other nutrient absorption. Fiber can also help with bile salt reabsorption from the small intestine, which is another component that contributes to lower cholesterol levels [25].

### 3.2. Protection of the Host’s GI Tract

The mucus layer in the GI tract covers and protects the gut epithelium, keeping pathogens away from the mucosa [39]. One of the host’s defenses against microbial invasions and infection is maintaining a well-structured and undamaged mucus layer. The gut microbiota and diet have been found to be two essential contributors to maintaining normal intestinal mucus structure and production [39]. Dietary fibers can promote mucus secretion by mechanically stimulating the intestinal epithelium [85]. Dietary fibers and SCFAs (such as acetate and butyrate) also boosts mucus synthesis and secretion by increasing goblet cell differentiation and mucin gene expression [39].

On the other hand, a low-fiber diet alters the gut microbiota. It promotes severe mucus layer degeneration, increasing susceptibility to infections and the development of chronic inflammatory conditions [92,93]. Low dietary fiber intake reduces microbial diversity and SCFA generation. It changes gut microbial metabolism toward less favorable substrates, such as dietary and endogenously provided proteins and host mucins, which is eventually harmful to the host [92,93]. Reduced fiber fermentation or increased protein fermentation reduces total SCFAs and butyrate production and increases the production of potentially harmful metabolites derived from amino acid fermentation [94]. These compounds’ cytotoxic and pro-inflammatory properties lead to the development of chronic illnesses, including colorectal cancer [95]. The switch in metabolism explains many diseases associated with a low-fiber diet. It also explains why microbial diversity is reduced in both humans and mice on a low-fiber diet [39].

### 3.3. Individuals May React Differently to the Same Fiber Intake

Despite multiple discoveries regarding the significance of fiber in inflammatory bowel disease, investigations have also revealed that the response to dietary fiber in IBD is variable [96]. Some comprehensive studies have also found no association between fiber consumption and IBD [97,98]. A possible explanation for this variation is the prevalence of inter-individual heterogeneity in the gut microbiota [99]. This heterogeneity has been causally related to varying effects of dietary fiber on host metabolic phenotypes, suggesting that a one-size-fits-all fiber supplementation strategy to improve health is unlikely to generate consistent outcomes across individuals [99].

Lancaster and colleagues found a similar result in a small crossover experiment with 18 participants using different fiber compositions. The researchers discovered that responses to different fiber types were not consistent across the participants, inferring that each person’s microbiome may influence their responses [100].

These findings suggest a complex link that exists between the effects originating from fiber, host gut microbiome, and host metabolism.

## 4. Conclusions

A nutritious and balanced diet should include sufficient fiber from whole grains, fruits, and vegetables. Along with the global transition to a more westernized lifestyle, the amount of fiber in our foods has decreased dramatically, exposing us to various ailments due to the loss of fiber’s protective function. IBD is defined by chronic pain and persistent inflammation that affects people of all age groups. IBD can also increase the risk of CRC. With considerable research demonstrating the benefits and properties of dietary fiber, it is more vital than ever to review the health benefits of high fiber intake for IBD patients. We have provided evidence that support the roles of dietary fiber in managing IBD. A diet enriched with fiber can help minimize inflammation, modulate immune response, restore the gut microbiome and prevent CRC in IBD. It can also help enhance general body health. However, it should be noted that an individual’s metabolism and microbiome composition can influence the extent of response to dietary fiber intake and one size may not fit all in terms of dietary fiber benefits. Possible caveats of this review include lack of clinical evidence to directly implicate SCFAs (products of fiber fermentation), into resolution of inflammation or mucosal healing and therefore large scale prospective longitudinal studies are needed to fully realize SCFA’s potential in reducing inflammation in patients with IBD.

## Figures and Tables

**Figure 1 biomedicines-10-01242-f001:**
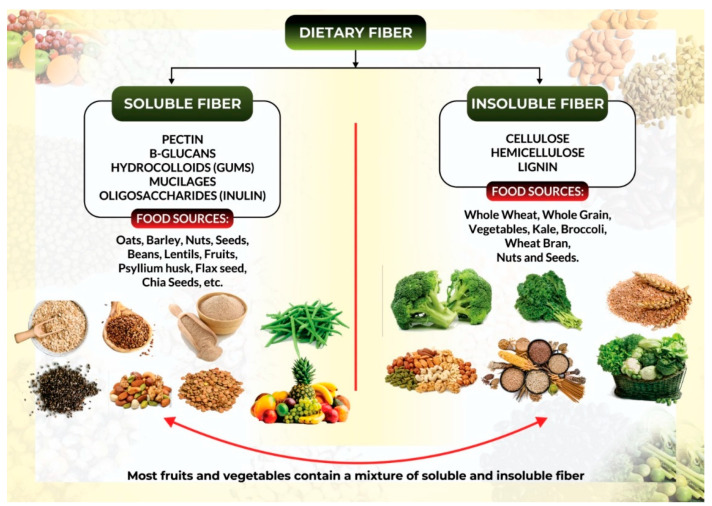
Classification of dietary fiber and its corresponding food sources.

**Figure 2 biomedicines-10-01242-f002:**
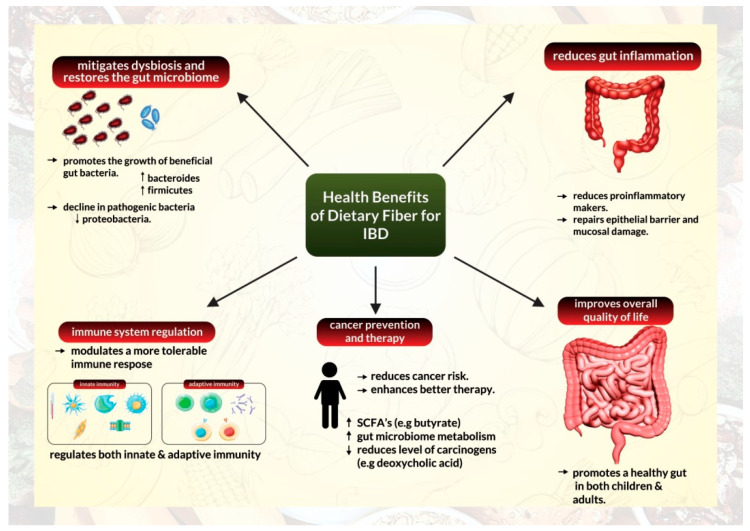
Health benefits of a high-fiber diet for management of inflammatory bowel disease.

## Data Availability

Not Applicable.

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
