# Peer review of "Health Benefits of Dietary Fiber for the Management of Inflammatory Bowel Disease"

_biomedicines, 2022, doi:10.3390/biomedicines10061242_

Round 1
Reviewer 1 Report
Dear authors,
I've read your manuscript "Health Benefits of Dietary Fiber for the Management of In-flammatory Bowel Disease". It was easy to read and follow. Unlike many other reviews about this topic, many discussed concepts were handled with a generalistic/minimalistic approach that, I think- makes your paper unable to stand out among many others.
My advice is to rebuild consistently the draft enriching it with more elaborated concepts, insights, and references.
The topic is, in fact, widely discussed and still considered "hot" no matter the the enormous amount of papers published over these last 20 years.
If you want to get scientists to read your review and get cited, I strongly advise making it more scientifically relevant for the community.
Reviewer 2 Report
Review article titled (Health Benefits of Dietary Fiber for the Management of In- flammatory Bowel Disease) by Yusuf et al. is an important and nicely written and organized one. I recommend the following revisions for improvement:
1- Figure legends : should be written under each figure.
2- Authors should make effort to organize the review by numbering and subnumbering, this will improve the whole appearance.
3- Title: In-flammatory should be "inflammatory"
4- Line 29: 7million need to be corrected.
5- Line 73: disease pathogenesis: mention pathogenesis of each type separately, CD or UC.
Reviewer 3 Report
Regarding the present manuscript, the following should be mentioned:
- The Introduction chapter contains too much data regarding the types of fibers and their description
- Data regarding the recommended fiber intake should be revised using more relevant references
- There is data regarding the connection between fiber consumption and cancer risk, which is beyond the purpose of this review
- There are significant and relevant references in the literature that have not been mentioned. Extensive literature research should be performed. Few E.g.: https://www.frontiersin.org/articles/10.3389/fped.2020.620189/full https://academic.oup.com/ecco-jcc/article/12/2/129/4372232 https://pubmed.ncbi.nlm.nih.gov/24445775/
- Conclusion could be adapted according to the literature findings
Round 2
Reviewer 1 Report
I've read the adjustments made to the manuscript and I can see a general improvement in the content, references and layout.
Reviewer 2 Report
thanks
Reviewer 3 Report
I have no further comments.